# Impact of Malnutrition Status on Muscle Parameter Changes over a 5-Year Follow-Up of Community-Dwelling Older Adults from the SarcoPhAge Cohort

**DOI:** 10.3390/nu13020407

**Published:** 2021-01-28

**Authors:** Laetitia Lengelé, Olivier Bruyère, Charlotte Beaudart, Jean-Yves Reginster, Médéa Locquet

**Affiliations:** 1WHO Collaborating Centre for Public Health Aspects of Musculoskeletal Health and Aging, Division of Public Health, Epidemiology and Health Economics, University of Liège, CHU—Sart Tilman, Quartier Hôpital, Avenue Hippocrate 13 (Bât. B23), 4000 Liège, Belgium; llengele@uliege.be (L.L.); c.beaudart@uliege.be (C.B.); jyr.ch@bluewin.ch (J.-Y.R.); medea.locquet@uliege.be (M.L.); 2Department of Sport Rehabilitation Sciences, University of Liège, 4000 Liège, Belgium; 3Physical, Rehabilitation Medicine and Sports Traumatology, SportS2, University Hospital of Liège, 4000 Liège, Belgium; 4Biochemistry Department, College of Science, King Saud University, Riyadh 11451, Saudi Arabia

**Keywords:** malnutrition, GLIM, SarcoPhAge, muscle mass, muscle strength, physical performance

## Abstract

This study aimed to assess the impact of malnutrition on the 5-year evolution of physical performance, muscle mass and muscle strength in participants from the SarcoPhAge cohort, consisting of community-dwelling older adults. The malnutrition status was assessed at baseline (T0) according to the “Global Leadership Initiatives on Malnutrition” (GLIM) criteria, and the muscle parameters were evaluated both at T0 and after five years of follow-up (T5). Lean mass, muscle strength and physical performance were assessed using dual X-ray absorptiometry, handgrip dynamometry, the short physical performance battery test and the timed up and go test, respectively. Differences in muscle outcomes according to nutritional status were tested using Student’s *t*-test. The association between malnutrition and the relative 5-year change in the muscle parameters was tested using multiple linear regressions adjusted for several covariates. A total of 411 participants (mean age of 72.3 ± 6.1 years, 56% women) were included. Of them, 96 individuals (23%) were diagnosed with malnutrition at baseline. Their muscle parameters were significantly lower than those of the well-nourished patients both at baseline and after five years of follow-up (all *p*-values < 0.05), except for muscle strength in women at T5, which was not significantly lower in the presence of malnutrition. However, the 5-year changes in muscle parameters of malnourished individuals were not significantly different than those of well-nourished individuals (all *p*-values > 0.05).

## 1. Introduction

Physiological changes occur with aging, increasing the risk of poor nutritional status [1]. Indeed, older adults are subjected to a decline in the senses of smell and taste and poor oral health, leading to a loss of appetite and anorexia [1,2]. Polypharmacy, inflammation and gastrointestinal disorders, which can lead to malabsorption, can also negatively impact nutritional status [1,2]. Alongside these physiological changes, psychosocial and environmental changes, namely, loneliness, depression, dementia, anxiety and loss of independence, may also be significant risk factors for malnutrition [1,2].

Malnutrition represents a major public health concern because of its high prevalence in the geriatric population and the costs it entails. Its prevalence rate varies widely according to the assessment tool, the geriatric setting studied or the definition criteria used. However, it is well recognized that a sizeable proportion of older adults may be malnourished, with a pooled estimated malnutrition risk prevalence of 22.6% in Europe [3]. Interestingly, in that meta-analysis using 22 malnutrition screening tools validated for use in older adults, the prevalence of malnutrition risk ranged from 8.5% in the community to 28% in the hospital [3]. Regarding the costs, malnutrition results in an economic burden for both the public health system and malnourished older individuals. Indeed, in a systematic review by Abizanda et al., malnutrition was associated with higher use of health care resources compared to well-nourished individuals, both in institutionalized or community-dwelling older adults, mainly due to higher general practitioner consultations, health monitoring and treatments [4].

Until recently, malnourished older adults were frequently under-diagnosed because of a lack of consensus on the assessment of this syndrome [1]. The first international definition was launched in 2015 by the European Society of Clinical Nutrition and Metabolism (ESPEN) following the action plan towards an optimal ageing described in the World Report on Ageing and Health [1]. In 2019, a new international agreement regarding the malnutrition diagnosis criteria was achieved by the worldwide clinical nutrition community, including the ESPEN society, to develop the Global Leadership Initiative on Malnutrition (GLIM) [5]. Previous studies have suggested that there is a risk of underestimating the malnutrition prevalence when the ESPEN definition is used [6,7]. Indeed, this definition requires the association of a weight loss with either a low Body Mass Index (BMI) or a decrease in muscle mass, while the GLIM phenotypic criteria require only one of the three factors. It is now well established that each criterion has to be considered separately given the fact that each of them may be indicative of malnutrition. The GLIM criteria revise and update, therefore, the previous international malnutrition definition, being partially based on it [8]. These criteria are meant to be applied across global health care sectors with the objective of standardizing clinical practice and research. Their development is an important step for the diagnosis of malnutrition in older people. Indeed, malnutrition has been shown to be associated with increased morbidity and mortality rates [9,10,11] and impaired quality of life [12], such as loss of independence in activities of daily living [13].

Interestingly, malnutrition seems to be related to muscle health decline and may contribute to the development of sarcopenia [14,15,16] and aggravate the age-associated loss of muscle mass, muscle strength and physical performance [17,18]. In one of our previous studies using observations from the SarcoPhAge cohort, we found that malnourished older individuals had a more than three-fold higher risk of developing sarcopenia after four years of follow-up than older individuals who were not malnourished [19]. There is growing evidence that links nutrition to muscle parameters [20]; however, how malnutrition affects the specific sarcopenia criteria decline remains unknown. Investigating the decline of the different muscle components in malnourished participants appears important in order to develop more effectively targeted therapeutic interventions, subsequently resulting in the prevention or slow-down of the related muscle health decline. Consequently, this study aimed to evaluate the impact of malnutrition, diagnosed according to the GLIM criteria, on changes in muscle mass, muscle strength and physical performance over a five-year period in older community-dwelling individuals.

## 2. Materials and Methods

This is a post-hoc analysis, as the primary objective of the SarcoPhAge cohort was the study of sarcopenia. Nevertheless, all the necessary data and measurements for the present study were already described in the research protocol in 2013.

### 2.1. Study Population

Participants from the SarcoPhAge (for “Sarcopenia and Physical Impairments with advancing age”) cohort were included in the present analysis. The detailed methodology of the SarcoPhAge study has already been described in detail elsewhere [21]. Briefly, the SarcoPhAge cohort is a Belgian population-based cohort developed in Liège (Belgium) in 2013. A total of 534 community-dwelling participants aged 65 years and older were recruited from press advertisements and general, geriatric, osteoporosis, rehabilitation, and rheumatology departments from an outpatient clinic in Liège, Belgium.

No specific exclusion criteria related to health or demographic characteristics were applied, except the exclusion of individuals with an amputated limb or with a BMI above 50 kg/m^2^, which was required for dual X-ray absorptiometry. The study was approved by the Ethics Committee of the Teaching Hospital of the University of Liege (reference 2012/277), with two amendments in 2015 and 2018. All volunteers gave their written informed consent.

The participants were followed up annually, and a clinical research assistant performed physical examinations and heath questionnaires to gather sociodemographic and anamnestic data. 

### 2.2. Diagnosis of Malnutrition

The diagnosis of malnutrition according to the Global Leadership Initiative on Malnutrition (GLIM) criteria was performed at baseline [5]. The GLIM algorithm implies a two-step approach with, first, screening using one of the validated tools proposed and, second, a diagnosis based on three phenotypic and two etiological criteria. The phenotypic criteria are used further to grade the severity of malnutrition, with specific threshold values different than those for the diagnosis. Since we had the data needed to diagnose malnutrition in all the included participants, the diagnosis could be established for the whole population. Therefore, we did not apply the screening part of the definition, useful and required in clinical routine for early identification of malnutrition. Malnutrition severity was not graded either, because the presence of malnutrition was used as a dichotomous variable for the objectives of the present study.

The GLIM malnutrition diagnosis requires at least one phenotypic and one etiological criterion that meet predefined thresholds. Based on the guidance proposed by the GLIM core leadership committee, the following thresholds were applied in the present study:-The phenotypic assessment includes (1) an unintentional weight loss higher than 4.5 kg in the past year [22], (2) a body mass index under 20 kg/m^2^ in participants younger than 70 years old or 22 kg/m^2^ in those older than 70 years old [5], and (3) a reduced muscle mass with a fat-free mass index (FFMI) under 17 kg/m^2^ in men and 15 kg/m^2^ in women or an appendicular lean mass index (ALMI) under 7 kg/m^2^ in men and 5.5 kg/m^2^ in women [5,23].-The etiological assessment involves (1) a reduced food intake determined according to the first item of the Mini Nutritional Assessment Short-Form (moderate or severe loss of appetite in the past three months) [24] and (2) inflammation evaluated by interleukin-6 (IL-6) and insulin-like growth factor 1 (IGF-1) [25], where the highest or the lowest quartile for IL-6 and IGF1, respectively, calculated in our own data set in both sexes, was considered a sex-specific threshold (i.e., IGF-1 ≤88 ng/mL in men and ≤82 ng/mL in women and IL-6 >3.84 pg/mL in men and >2.99 pg/mL in women). Inflammation is highlighted once the value of IL-6 is above or IGF-1 is below these thresholds. These thresholds are similar to other previous published thresholds for community-dwelling older adults [26,27]. The biomarkers used in the present study were identified as relevant for geroscience-guided clinical trials, robust, with a consistent ability to predict clinical and functional outcomes, responsive to intervention, and with a reliable and feasible measurement according to a comprehensive review conducted by a panel of experts [25]. From all the biomarkers considered in this review, IL-6 was selected over CRP for the assessment of inflammation because it was more robust and considered as more appropriate to reflect the aging process. TNF-α was not selected for this present study because it tends to be low and unstable when stored at a temperature of −80 °C. Regarding IGF1, this biomarker was selected for its responsiveness to caloric restriction given the fact that it was used for the diagnosis of malnutrition.

### 2.3. Muscle Parameters

To investigate the muscle health of our participants, three measurements were performed: muscle mass, muscle strength and physical performance.

Physical performance was evaluated using the Short Physical Performance Battery (SPPB) test, which consisted of three physical assessments: balance, 4-m walking speed, and the chair stand test [28]. A score between 0 and 4 points was assigned for each of the three separate tests, with a maximum of 12 points. Physical performance was also assessed using the timed up and go (TUG) test, where participants were asked to stand up from a chair, walk three meters, turn around, return, and sit down again [29]. The performance was timed and reported in seconds.

Muscle mass was estimated with a dual energy X-ray absorptiometer (DXA) (Hologic Discovery A, USA), calibrated daily. Fat-free mass, that is, the subtraction of fat mass from total body mass, and appendicular lean mass, the sum of the muscle mass in both arms and legs, were obtained from whole-body DXA scans and were divided by height squared (kg/m^2^) to obtain the fat-free mass index (FFMI) and the appendicular lean mass index (ALMI) values, respectively.

Muscle strength was measured with a handgrip hand-held dynamometer (Saehan Corporation, MSD Europe Bvba, Brussels, Belgium), calibrated yearly. We followed standardized procedures by asking participants to squeeze as hard as possible three times per hand. The highest value of the six measurements, in kg, was considered [30].

Cut-off values were used to highlight impaired muscle parameters:Low physical performance with an SPPB score ≤ 8 points [8] and a TUG score ≥ 20 s [23].Low muscle mass with an FFMI under 17 kg/m^2^ in men and 15 kg/m^2^ in women or an ALMI under 7 kg/m^2^ and 5.5 kg/m^2^ in men and women, respectively [23].Low muscle strength with grip strength under 27 kg and 16 kg in men and women, respectively [23].

### 2.4. Confounding Factors

The following variables that could potentially impact muscle health and nutritional status according to the literature [19,31,32,33,34,35,36,37] were considered covariates: age, sex, smoking status (yes/no), alcohol consumption (yes/no), number of comorbidities per individual, number of drugs consumed per individual, cognitive status assessed by the Mini-Mental State Examination (MMSE) [38], self-reported physical activity level with the Minnesota Leisure Time Activity Questionnaire [39], and the baseline value of the muscle component.

### 2.5. Statistical Analysis

The distribution of the continuous variables was evaluated by the difference between the mean and the median values, histogram, quantile-quantile plot, and the Shapiro-Wilk test. Nevertheless, continuous variables had to be treated and reported following a Gaussian distribution because of the predictive mean matching method executed in the multiple imputations procedure [40]. The continuous variables were then expressed as the mean ± standard deviation, and categorical variables were reported in absolute (N) and relative frequencies (%).

The prevalence of malnutrition according to the GLIM definition was assessed. Participants’ baseline characteristics and muscle parameters at baseline and after five years of follow-up were compared between malnourished and non-malnourished individuals using Student’s *t*-test for quantitative variables and using the chi-squared test for qualitative variables.

The changes in muscle mass (FFMI and ALMI), muscle strength (grip strength), and physical performance (SPPB and TUG tests) were calculated by the relative change (i.e., difference between the baseline and follow-up values, as a percentage of the baseline value) from baseline to the fifth year. Relative changes allowed us to consider the effect, on the 5-year change on the potential large imbalance in the baseline muscle parameters according to the diagnosis of malnutrition. First, a crude model was performed to compare the mean relative changes in the muscle parameters according to malnutrition status using Student’s *t*-test for independent data. Then, the association between the relative change between T0 and T5 of these muscle parameters and malnutrition was determined through multiple linear regressions with one of the muscle parameters as a dependent variable and nutritional status as an independent variable, adjusted for confounding factors using the enter method. A first multivariate analysis model was adjusted for age only for muscle mass and muscle strength, as the analyses were separated for men and women and adjusted for age and gender for physical performance. A second model was then performed, adjusted for all the potential covariates identified above. The baseline value of the muscle component was also used as a covariate in the fully adjusted model, as we assumed baseline muscle parameter imbalance according to the diagnosis of malnutrition, which could influence the possible change range of the muscle parameters during the follow-up. The individuals with lower baseline values could undergo a larger improvement, and inversely. This phenomenon, known as regression to the mean, can be taken into account, as well as the ceiling and floor effects, with this adjustment [41].

The proportion of individuals presenting low muscle parameters according to the defined thresholds was estimated at baseline and at the fifth year of follow-up. We were then able to compare, first, the proportion of individuals below the cut-off values in malnourished and non-malnourished individuals, both at baseline and at the five-year follow-up, using the chi-squared test. Second, we calculated the change in the proportion of individuals with low muscle parameters between baseline and the fifth year of follow-up and compared these changes in proportion according to nutritional status with the two proportions Z test for independent samples.

Missing data were handled using multiple imputations according to the Markov chain Monte Carlo model, in which the predictive mean matching method was applied for continuous variables. Five datasets were computed, and pooled estimates from these datasets were used to report the analysis results.

The results were considered statistically significant at a degree of uncertainty of 5% (*p* < 0.05). The SPSS Statistics 24 (IBM Corporation, Armonk, NY, USA) software package was used for the analyses.

## 3. Results

The baseline characteristics of the participants are presented in Table 1. Out of the 534 participants recruited initially in the SarcoPhAge cohort, data needed to diagnose malnutrition at baseline, according to the GLIM criteria, were available for a total of 411 participants (mean age of 72.3 ± 6.1 years, 56% women). Among these participants, 96 (23.3%) were diagnosed with malnutrition at baseline. The malnourished participants were significantly more women (*p*-value < 0.05) and smokers (*p*-value < 0.01), had a lower body mass index (23.9 ± 4.0 kg/m^2^ versus 27.7 ± 4.5 kg/m^2^, *p*-value < 0.001), had more concomitant diseases (5.0 ± 2.4 versus 4.0 ± 2.3, *p*-value < 0.001), and had a worse cognitive status (27.6 ± 2.3 versus 28.2 ± 2.0 out of 30 points on the MMSE scale, *p*-value = 0.01) than the well-nourished participants.

When looking at the muscle parameters according to nutritional status (Table 2), the malnourished individuals had a significantly lower SPPB test score, took more time to perform the TUG test, had less fat-free mass and appendicular lean mass, and had a lower muscle strength compared to well-nourished individuals, in both men and women and at baseline but also at the end of the five years of follow-up (all *p*-values being under 0.05). The only exception was that muscle strength in malnourished women was not significantly lower than that in well-nourished women at the five-year follow-up (*p*-value = 0.99).

The relative changes over five years are presented in Table 3 for muscle mass and muscle strength and in Table 4 for physical performance. In the univariate analyses, the results showed a significant difference between well-nourished and malnourished older adults in the relative change over five years of the FFMI and the ALMI only, both in men and women (all *p*-values < 0.05). These results remained significant in the first adjusted model including age as a covariate (all *p*-values < 0.05) but not in the fully adjusted model. Indeed, when adjusted for all covariates, the relative changes in all muscle parameters (namely, FFMI, ALMI, muscle strength, SPPB and the TUG test) over five years of follow-up did not differ significantly according to nutritional status (*p*-value > 0.05). 

The variability of the dependent variables (i.e., muscle parameters) that was explained by the independent variables included in the models, which is represented by the R^2^, was at least three times higher in the second model compared to the first for each outcome assessed. Furthermore, when we explored the effect of each confounding variable on the dependent variable (i.e., the muscle parameter), the baseline value of the parameter was the only significant variable in the model for all the muscle parameters. This means that this confounding variable included in the second model explains, at least partially, the non-significant changes observed between malnourished and well-nourished individuals. 

The proportion of individuals with muscle parameters below the threshold, presented in Table 5, was significantly higher in the malnutrition group at T0 and T5 (*p*-values < 0.05). Only the proportion of individuals with low muscle strength was not significantly different at the fifth year of follow-up according to nutritional status (55.0% vs. 45.2%, *p* = 0.11).

Globally, the number of individuals below the threshold values was lower at the five-year follow-up than at baseline, both in malnourished and well-nourished individuals, except for muscle strength. For this last parameter, an important increase in the proportion of older individuals reaching this cut-off was observed after five years of follow-up. Regarding the change in the prevalence of subjects below the threshold between T0 and T5, the difference was non-significant according to malnutrition status, except for the FFMI and ALMI (*p*-values < 0.05).

## 4. Discussion

In our analyses, malnourished individuals had globally weaker muscle mass, muscle strength and physical performance than well-nourished individuals, both at baseline and at five years follow-up. Similar conclusions are drawn when the thresholds of low muscle parameters are applied, as the proportion of individuals reaching these thresholds is more important in malnourished than in well-nourished individuals at the two same time points. However, the evolution of the muscle parameters after five years did not seem to be impacted by malnutrition in the present study. Indeed, the five-year changes in all the muscle parameters were not significantly different according to nutritional status after adjustment for covariates. Regarding the change in the prevalence of individuals below thresholds between T0 and T5, the difference was non-significant according to malnutrition status except for muscle mass (i.e., FFMI and ALMI), in which an improvement was seen in the number of individuals reaching these thresholds in the malnourished participants compared to the well-nourished participants (*p* < 0.05).

The cross-sectional association between malnutrition and muscle health, as observed in the present study, has already been identified in other studies using different malnutrition criteria than the GLIM ones, the GLIM definition being very recent. In the study of Liguori et al., similar results to ours were observed for muscle mass and strength, which were negatively associated with the risk of malnutrition, assessed using the mini-nutritional assessment (MNA) questionnaire, in non-institutionalized older adults [42]. Moreover, a negative association between physical performance and malnutrition was found in the research of Ramsey et al., in which malnourished geriatric outpatients, assessed by the Short Nutritional Assessment Questionnaire (SNAQ), had lower SPPB and higher TUG scores than well-nourished patients (*p* < 0.05) [43]. However, no significant association between malnutrition and handgrip strength was found in this study, as opposed to our results and those of Liguori et al. The divergence regarding muscle strength can potentially be explained by the use of different malnutrition criteria. Indeed, the MNA and the SNAQ questionnaires are recommended for the screening part of the definition, a first step before the diagnosis of malnutrition assessed by the GLIM criteria in the present study. Furthermore, it has been shown that dissimilar criteria could identify different patients in the same population [44] and that the MNA itself may overestimate the risk of malnutrition in geriatric rehabilitation settings [45] and in hospitalized older adults [46]. Interestingly, the GLIM definition itself includes low muscle mass as a phenotypic criterion. It seems therefore coherent that individuals diagnosed with malnutrition had lower muscle health in our study. The cross-sectional link between nutrition and muscle health is well established because nutritional strategies are now an integral part of the prevention and management of sarcopenia [47].

Regarding the impact of malnutrition on the five-year change in muscle parameters, we are the first, to our knowledge, to have investigated it with the GLIM criteria in community-dwelling older adults. In a previous study using data from our SarcoPhAge cohort, individuals diagnosed with malnutrition by the GLIM criteria, had a three-fold increased risk of becoming sarcopenic compared with well-nourished individuals after four years of follow-up [19]. Therefore, this study aimed to explore these results further by analyzing the impact of malnutrition on each specific muscle parameter. The results of the present study revealed, surprisingly, that the relative changes in muscle mass, muscle strength and physical performance did not differ according to nutritional status. We can presume that the possible range in decline of muscle parameters was smaller, and muscle parameters were subjected to floor effects [48], especially as observed for muscle strength at five-year of follow-up in women. In fact, malnourished individuals had lower muscle baseline values than well-nourished individuals. Similarly, in a study by Goodpaster et al., examining the three-year changes in knee extensor strength in older adults, it was found, as in the present study, that those who were stronger at baseline were more likely to lose more strength during follow-up, whether change data were expressed in absolute or proportionate changes, and this phenomenon was observed for both men and women [49]. In the present study, the baseline value of the muscle parameter was included in the fully adjusted model. When we explored the effect of each covariate on the dependent variable (i.e., the muscle parameter) in the model, the baseline value of the parameter appeared to be significant for all the muscle parameters. This means that the baseline value of the muscle parameters explains, at least partially, the non-significant changes observed between malnourished and well-nourished individuals and supports the hypothesis developed above.

The number of older adults below the thresholds of muscle mass and physical performance has decreased in five years, especially in malnourished patients, which is quite unexpected. Nevertheless, the mean of these parameters in malnourished patients at baseline was close to the threshold values. Therefore, we can assume that the transition towards and away from the low muscle parameter group could have been made even with a small change, potentially non-clinically pertinent. In other words, this result highlights the relevance of measuring the minimal clinically important difference (MCID), instead of threshold values, when evaluating evolution over time. Indeed, even if fewer older adults had a low muscle mass or physical performance at five years of follow-up, this does not mean that the improvement observed is relevant, for the patients or clinically. Currently, the MCID for muscle strength, the SPPB or the TUG test can be found in the literature [48], but it is still not well defined in older adults and clinical practice, as the values can vary widely according to the calculation method used or the population targeted. Furthermore, the MCID is context-dependent, is influenced by the health state of the patients and their demographic characteristics [50] and is baseline value-dependent [51], as patients with lower scores at inclusion required more change to report a meaningful difference. Therefore, we could not apply MCID values in this study, as additional research is needed to harmonize the recommendations. Nevertheless, this could provide considerable information regarding changes in muscle parameters. From a statistical point of view, the changes appeared to be non-significant according to nutritional status, but from the patients’ perspective, these changes could potentially have dissimilar repercussions on their global health status or quality of life in the presence or absence of malnutrition.

While there did not appear to be any major differences in the number of subjects reaching the threshold values regarding muscle mass and physical performance after five years of follow-up, this was not the case for muscle strength. The number of patients below the cut-offs more than doubled after five years, which is probably the leading cause of the increased incidence of sarcopenia observed in our previous study [19]. Muscle strength is also the parameter that underwent the most important relative difference of all the muscle parameters. First, muscle strength was the furthest parameter from the threshold value at baseline compared to muscle mass or physical performance. Therefore, we can assume that the possible range of decrease was higher for muscle strength. Second, this parameter has already been identified as a responsive component of muscle health, as the age-associated decline in muscle strength is more rapid than muscle mass loss, with an annual rate three to five times higher than muscle mass in healthy older adults [49,52]. Moreover, muscle strength is now considered the principal determinant of sarcopenia, as it is recognized as the best parameter in predicting adverse outcomes [23]. Finally, contrary to our expectations and the literature [2,18,53,54], muscle mass underwent a limited decrease, even an increase in the ALMI in women, and physical performance improved in our analyses after five years of follow-up. In a previous study of the SarcoPhAge cohort [55], we noticed a significant increase in the level of physical activity after three years of follow-up. The increase in physical activity level, which is associated with better muscle health [56,57,58,59], could explain, to a certain extent, the evolution observed for these two muscle parameters in the present study.

### Strengths and Limitations

The present study has several limitations that can be underlined. First, the constitution of our population has potentially brought a selection bias, as they are recruited volunteers. Furthermore, the malnutrition prevalence of 23.4% in our cohort appeared to be higher than the prevalence of 10.7% and 14% found in other studies using the GLIM criteria in community-dwelling older adults [60,61]. Consequently, our results could be limited in their external validity. Second, the length of follow-up, which was five years, was probably not long enough to highlight an evolution significantly different according to nutritional status, especially for muscle mass. Last, nutrition is not the only factor that can impact muscle health. Resistance and aerobic exercise training are also known to play a positive role in the management of muscle health decline [62] by enhancing myofibrillar protein synthesis [63]. The analyses were adjusted for physical activity level, but we did not adjust for the type of physical activity because we did not have the data. The participants could also have benefited from a nutritional intervention during the five-year follow-up, such as nutritional supplementation, and this could have impacted the outcomes measure by improving the muscle parameters or, at least, slowing their decrease [64,65]. Unfortunately, this potential confounding variable was not assessed in this cohort. 

Regarding the criteria and threshold values used in the present study to diagnose malnutrition, these were chosen according to the recommended propositions from the GLIM consensus, except for the unintentional weight loss criteria from the FRIED questionnaire, as we did not have any other available data. However, details of how to categorize each criteria are not yet provided, as evidence on clear threshold values are lacking [66]. For example, the GLIM consensus report indicates that the thresholds for the “reduced food intake” and “weight loss” criteria are widely reported in the literature and vary according to the malnutrition tool used [5]. This is also the case for inflammation, for which clear measurement methods or threshold values are not communicated in the GLIM definition. This could lead to variations in the proportion of malnourished individuals across studies, e.g., a prevalence of 10.7% found in a study of Yeung et al., applying different measures and thresholds for the GLIM criteria [60] than in the present study. Nevertheless, this highlights the need for a consensus regarding the assessment of the different malnutrition criteria and brings new research perspectives to identify the thresholds that are most predictive of adverse outcomes. Regarding the strengths of this study, multiple imputations were carried out to handle missing data. Nevertheless, sensibility analyses performed without imputations on data available from 224 participants assessed at T5 showed similar results (data not shown, available on request). This assesses the robustness of our present findings. In addition, this study produced original results by being the first to investigate the impact of malnutrition, diagnosed with the GLIM definition, on five-year changes in muscle mass, muscle strength and physical performance. Therefore, this study brings new elements to the management of muscle health decline and its relationships with malnutrition. Other studies are therefore required to confirm our findings. Some perspectives could also be interesting in future studies: investigating the impact of severe malnutrition on muscle health and adopting a dynamic approach by measuring the cumulative incidence of malnutrition or the transition of the participants from malnourished to well-nourished status (i.e., the natural course of the disease) during the follow-up period would allow a better understanding of the complex mechanisms of malnutrition and its relationship with health decline.

## 5. Conclusions

In conclusion, individuals diagnosed with malnutrition had weaker muscle mass, muscle strength and physical performance than well-nourished individuals in the cross-sectional analyses. However, after five years, there did not seem to be any difference in the evolution of the muscle parameters according to nutritional status. Furthermore, in the present study, muscle strength decline seemed to be the major leading cause of the incidence of sarcopenia in the SarcoPhAge cohort.

## Figures and Tables

**Table 1 nutrients-13-00407-t001:** Baseline characteristics of participants in the SarcoPhAge study (*n* = 411).

	Total Study Sample (*n* = 411)	Malnutrition		*p*-Value ^b^
		Yes (*n* = 96)	No (*n* = 315)	
Age, years	73.2 ± 6.1	73.9 ± 6.8	73.0 ± 5.8	0.19
Gender				0.046
Men	182 (44.8)	34 (35.4)	148 (47.0)	
Women	229 (55.7)	62 (64.6)	167 (53.0)	
Smoking status, yes	33 (8.0)	14 (14.6)	19 (6.0)	0.007
Alcohol consumption, yes	215 (52.3)	54 (56.3)	161 (51.1)	0.38
Body mass index, kg/m^2^	26.8 ± 4.7	23.9 ± 4.0	27.7 ± 4.5	<0.001
Number of concomitant diseases per individual	4.2 ± 2.4	5.0 ± 2.4	4.0 ± 2.3	<0.001
Number of drugs per individual	5.8 ± 3.4	6.3 ± 3.5	5.6 ± 3.4	0.059
Mini-Mental State Examination (MMSE), 30 points	28.1 ± 2.1	27.6 ± 2.3	28.2 ± 2.0	0.012
Level of physical activity, kcal/day ^a^	1102.9 ± 1257.8	1057.3 ± 1267.9	1116.7 ± 1256.4	0.68

Quantitative variables were expressed as mean (standard deviation), and binary or categorical variables were described by absolute (*n*) and relative (%) frequencies. ^a^ This variable did not follow a Gaussian distribution. but we had reported it as following a Gaussian distribution because of the predictive mean matching method executed in the multiple imputations procedure. ^b^
*p*-values obtained from a Student’s *t*-test for quantitative variables and a chi-squared test for qualitative variables.

**Table 2 nutrients-13-00407-t002:** Muscle parameters at baseline and at five-year follow-up according to nutritional status (*n* = 411).

	Baseline			Five-Year Follow-Up		
	Malnourished (*n* = 96)	Well-Nourished (*n* = 315)	*p-*Value *	Malnourished (*n* = 96)	Well-Nourished (*n* = 315)	*p-*Value *
Fat-free mass index (FFMI), kg/m^2^						
Men	16.8 ± 2.2	19.3 ± 2.3	<0.001	16.7 ± 2.1	18.4 ± 2.2	<0.001
Women	14.3 ± 1.4	15.8 ± 1.9	<0.001	14.9 ± 1.9	15.9 ± 1.8	0.002
Appendicular lean mass index (ALMI), kg/m^2^						
Men	7.0 ± 1.0	8.2 ± 1.0	<0.001	6.9 ± 1.0	7.6 ± 1.0	0.005
Women	5.5 ± 0.7	6.3 ± 1.0	<0.001	5.9 ± 0.9	6.3 ± 0.8	0.002
Grip strength, kg						
Men	32.5 ± 10.1	40.8 ± 7.8	<0.001	22.6 ± 9.5	30.7 ± 9.0	0.045
Women	19.8 ± 5.4	22.6 ± 7.1	0.005	14.7 ± 6.3	14.7 ± 6.5	0.993
Short Physical Performance Battery (SPPB), /12 Points	8.4 ± 2.8	9.7 ± 2.0	<0.001	8.8 ± 2.5	9.9 ± 1.9	0.001
Timed up and go (TUG), s	13.4 ± 7.3	11.0 ± 4.9	<0.001	12.4 ± 6.5	9.9 ± 4.0	0.001

* *p*-values obtained from a Student’s *t*-test.

**Table 3 nutrients-13-00407-t003:** Relative change over time of muscle mass and muscle strength over a five-year period according to nutritional status and gender (*n* = 411).

		Crude Model		Model 1	Model 2
		Relative Change T0–T5 (%)	*p*-Value *	*p*-Value **	*p*-Value ***
Men (*n* = 182)	FFMI, kg/m^2^				
	Well nourished Malnourished	−4.3 ± 6.4 −0.1 ± 6.9	0.03	0.02	0.17
	ALMI, kg/m^2^				
	Well nourished Malnourished	−6.4 ± 8.4 −0.7 ± 9.6	0.04	0.03	0.19
	Grip strength, kg				
	Well nourished Malnourished	−24.1 ± 21.3 −23.5 ± 48.3	0.96	0.94	0.55
Women (*n* = 229)	FFMI, kg/m^2^				
	Well nourished Malnourished	−1.2 ± 7.1 4.6 ± 8.2	0.04	0.04	0.21
	ALMI, kg/m^2^				
	Well nourished Malnourished	0.5 ± 9.8 6.5 ± 11.4	0.005	0.005	0.30
	Grip strength, kg				
	Well nourished Malnourished	−33.2 ± 29.6 −26.8 ± 33.1	0.46	0.45	0.45

* *p*-values obtained from a Student’s *t*-test ** *p*-values obtained from multiple linear regression adjusted for age *** *p*-values obtained from multiple linear regression adjusted for age, smoking status, alcohol consumption, number of comorbidities, number of drugs consumed, MMSE, Physical activity level and baseline value of the muscular component.

**Table 4 nutrients-13-00407-t004:** Relative change over time of physical performance over a five-year period according to the nutritional status (*n* = 411).

	Crude Model		Model 1	Model 2
	Relative Change T0–T5 (%)	*p*-Value *	*p*-Value **	*p*-Value ***
SPPB, 12 Points				
Well nourished Malnourished	4.7 ± 1.7 15.8 ± 6.8	0.22	0.24	0.59
TUG, s				
Well nourished Malnourished	−6.0 ± 1.7 −1.9 ± 3.8	0.40	0.24	0.12

* *p*-values obtained from a Student’s *t*-test ** *p*-values obtained from multiple linear regression adjusted for age and gender *** *p*-values obtained from multiple linear regression adjusted for age, smoking status, alcohol consumption, number of comorbidities, number of drugs consumed, MMSE, Physical activity level and baseline value of the muscular component.

**Table 5 nutrients-13-00407-t005:** Low muscle mass, muscle strength and physical performance at T0 and T5 and the change in the proportion of individuals with low muscle parameters between T0 and T5 compared according to nutritional status (*n* = 411).

	Baseline, *n* (%)	*p*-Value	5-Year Follow-Up, *n* (%)	*p*-Value	Proportion Difference
Low FFMI					
Malnourished Well nourished	69 (71.9) 78 (24.8)	<0.001	59 (61.3) 80 (25.3)	<0.001	−10.6 * +0.5 *
Low ALMI					
Malnourished Well nourished	60 (62.5) 38 (12.1)	<0.001	43 (45.2) 58 (18.3)	0.004	−17.3 * +6.2 *
Low Grip strength					
Malnourished Well nourished	20 (20.8) 22 (7.0)	<0.001	53 (55.0) 126 (45.2)	0.108	+34.2 +38.2
Low SPPB					
Malnourished Well nourished	42 (43.8) 64 (20.3)	<0.001	37 (38.5) 55 (17.4)	0.01	−5.3 −2.9
Low TUG					
Malnourished Well nourished	12 (12.3) 14 (4.4)	0.005	9 (9.4) 6 (1.8)	0.025	−2.9 −2.6

* Indicates a significant change in the five-year prevalence of low muscle parameters between malnourished and well-nourished patients.

## Data Availability

The data presented in this study are available on request from the corresponding author. The data are not publicly available due to GDPR policies and restrictions, i.e., all information that could compromise the privacy of the participants.

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
