# Peer review of "Impact of Malnutrition Status on Muscle Parameter Changes over a 5-Year Follow-Up of Community-Dwelling Older Adults from the SarcoPhAge Cohort"

_nutrients, 2021, doi:10.3390/nu13020407_

Round 1

Reviewer 1 Report

The topic of the study is current, the paper is interestingly written and the findings are clearly presented. I would like to make the following suggestions for improvement in the method section: •As described in your study, in the SarcoPhAge cohort the MNA was conducted for baseline assessment. Therefore, the GLIM criteria were applied retrospectively to the patient collective (posthoc not prospectively) and not during the first examination (line 99-100). This could possibly have led to an inaccuracy in the composition of your subgroups. This limitation could have led to an overestimation of the proportion of malnourished test subjects, which was relatively high in your study (23%) and would be estimated at 10% in a population-based study. Did the MNA show a similar distribution of the subgroups? Why didn’t you use the MNA instead? •IL-6 is rarely used in clinical routine to detect chronical or acute inflammation. The limit value mentioned in the study also appears to be very low (> 3.84 pg/ml) to me as in our laboratories a mild inflammatory reaction is only assumed in a range of at least 12-15 pg/ml. You also define the "lowest quartile" (line 121) of IL-6 values as a group with increased inflammation, which should be reversed. •Why did you choose IGF-1 as an inflammatory marker and not CRP instead? Studies showed that IGF-1 inhibits the expression of inflammatory markers, although other studies concluded that IGF-1 has proinflammatory functions. Defining a lower and/or upper quartile from a patient group using a poorly used inflammatory biomarker does poorly define acute or chronical inflammation according to the GLIM criteria. The inclusion of patients with low inflammatory activity maybe let to an overestimation of the malnourished group. Major comments 1.I suggest redefining the malnourished patient group only using patients with either high inflammatory activity or a reduced food intake or assimilation disease burden. The patient group should also be subdivided in patients with risk of malnutrition, patients with moderate or severe malnutrition. This way a subgroup analysis could led to more significant differences between the well-nourished and malnourished patients. 2.Did you find significant changes between the groups at T0 and T5 using the MNA? Minor comments: •The level of physical activity (Table 1) shows a large standard deviation. Was the physical activity overestimated in patients? •Conversion of Table 3: FFMI, kg/m2 is written in bold and the group of men is underlined while the group of women is not underlined.

Reviewer 2 Report

The authors focus with this manuscript on the impact of malnutrition on the 5-year evolution of muscle performance, mass and strength in community-dwelling older adults, participant from a previous study already published from the same journal.

This highlights one really important concerning point, since malnutrition is associated to increased morbidity and mortality rates. Thereby we recognize the need of methods to foresee it and to prevent/delay its detrimental effects.

Malnutrition is assessed through the application of the new GLIM criteria.

My first question is why the authors do not compare this way of assessing the syndrome with other previous one? I can understand that probably the lack of other ways assessing it can make difficult the comparison, but the two examples reported in discussion (Liguori et al., or Ramsey et al., ) are using other two questionnaires (MNA and SNAQ) that are not even completely explained. I would add some more reflexions about it in the introduction, highlighting the differences introduced by using GLIM compared to other previous methods. While in discussion I would highlight the advantages of using GLIM, letting the reader more easily understand which can be the differences with previous questionnaires and possibly, the advantage of assessing malnutrition with GLIM criteria.

Considering GLIM criteria, I have a question over the differences in circulating levels of many growth factors and cytokines. Why the authors stress the attention on two of them (IGF1 and IL-6) while usually more could be assessed? From the reference they report, we know also TNFa (just to mention one) plays a pivotal role in inflammation. Any further explanation for their choice and possibly are there more data available on other pro-inflammatory cytokines or growth factors?

Moreover, usually the basal level of proinflammatory cytokines are used to increase in elderly. Why at lines 124-125 the authors talk about a decrease of the levels of IGF1 (expectable and reasonable) but also of IL-6 (that I would expect to increase)?

In the results, Table1 reports the data over women. It is normal to ask also the ones for men. Did I miss anything in the text regarding this? I still think it is needed to report them for the completeness of the study and I would add them.

In Table 2 the authors report for women, in the 5-years follow-up group, no differences in grip strength depending on the nutrition of the subjects. I would extend a bit more also in discussion any speculation on this: is it due to a decrease due to the age for the well-nourished or to the fact that there is a compensation/conservative mechanism just present in female malnourished?

In Table3 there are the two models for evaluating the significance of the differences. Any speculation about the differences we can notice just with one but not with the second? What are these differences due to? How could we think to use one or better the other model?

Finally, I would represent the data in a more graphical way to help the reader to focus on the differences that also the authors stress on.

Reviewer 3 Report

Line 52: Correct name: Abizanda

As ALMI is something more precise than FFMI for diagnose of poor muscle mass, could you specify the number of people submitted to each determination.

- I suppose there were people previously diagnosed of malnutrition. It have been keep in mind nutrition therapy as confounding factor for muscle and malnutrition evolution? Describe it. If not, add it in the paper limitations.

You should underline in discussion that the parameters used in your study for establishing GLIM criteria for malnutrition are not exactly what they recommend.. You use unintentional weight loss higher than 4.5 kg in the past year, instead of percentage of weight loss in the last 6 months; Reduce food intake determined by moderate or severe loss of appetite in the past 3 months (according to the first item of the MNA-SF), instead of reduction of food intake in the last 1-2 weeks; inflammation parameters… This is an important aspect to compare percentages of malnutrition by GLIM criteria with other series with similar characteristics.

Round 2

Reviewer 1 Report

Many thanks to the authors for the detailed response to the revision.

„The biomarkers used in the present study were identified as relevant for geroscience-guided clinical trials, robust, with a consistent ability to predict clinical and functional outcomes, responsive to intervention, and with a reliable and feasible measurement according to a comprehensive review conducted by a panel of experts (Justice et al., 2018).“

According to the review, the TAME study (https://doi.org/10.1007/s11357-018-0042-y) was conducted to indentify biomarkers correlated to age-related multi-morbidity and functional decline and not to investigate clinical inflammation („Next selection criteria were derived and applied to refine this set emphasizing:(1) measurement reliability and feasibility; (2) relevanceto aging; (3) robust and consistent ability to predict all-cause mortality, clinical and functional outcomes; and(4) responsiveness to intervention.“).

The cited article is about biomarkers associated with inflamm-aging and does not investigate biomarkers which should be used to indicate acute or chronic clinical inflammation. To my understanding, the GLIM critieria (https://doi.org/10.1016/j.clnu.2018.08.002) refer to infection-related and disease-related inflammation (disease burden / inflammatory condition) in their etiologic criteria and not to subclinically increased inflammatory biomarker.

„For example, major infections, burns, trauma, and closed head injury are associated with acute inflammation of a severe degree. Indicators of inflammation may include fever, negative nitrogen balance, and elevated resting energy expenditure. […] Supportive proxy measures of inflammation can include laboratory indicators like serum C-reactive protein (CRP), albumin, or pre-albumin.“ (Cederholm et al.)

As a limitation of the study it should be mentioned that IL-6 and IGF-1 reflect a minimal or no perceived inflammation, which could have led to a higher prevalence of malnutrition in the patient group.

Reviewer 2 Report

The authors addressed the points highlighted during the revision.

Author Response

"The authors addressed the points highlighted during the revision."

We thank the reviewer for the highly relevant suggestions discussed for the review.